# SLC26A9 as a Potential Modifier and Therapeutic Target in Cystic Fibrosis Lung Disease

**DOI:** 10.3390/biom12020202

**Published:** 2022-01-25

**Authors:** Giulia Gorrieri, Federico Zara, Paolo Scudieri

**Affiliations:** 1Department of Neurosciences, Rehabilitation, Ophthalmology, Genetics, Maternal and Child Health (DiNOGMI), University of Genoa, 16132 Genoa, Italy; giulia.gorrieri@edu.unige.it (G.G.); federico.zara@unige.it (F.Z.); 2Medical Genetics Unit, IRCSS Giannina Gaslini Institute, 16147 Genoa, Italy

**Keywords:** SLC26A9, cystic fibrosis, gene modifiers, anion transport, lung physiology, airway epithelium

## Abstract

SLC26A9 belongs to the solute carrier family 26 (SLC26), which comprises membrane proteins involved in ion transport mechanisms. On the basis of different preliminary findings, including the phenotype of SlC26A9-deficient mice and its possible role as a gene modifier of the human phenotype and treatment response, SLC26A9 has emerged as one of the most interesting alternative targets for the treatment of cystic fibrosis (CF). However, despite relevant clues, some open issues and controversies remain. The lack of specific pharmacological modulators, the elusive expression reported in the airways, and its complex relationships with CFTR and the CF phenotype prevent us from conclusively understanding the contribution of SLC26A9 in human lung physiology and its real potential as a therapeutic target in CF. In this review, we summarized the various studies dealing with SLC26A9 expression, molecular structure, and function as an anion channel or transporter; its interaction and functional relationships with CFTR; and its role as a gene modifier and tried to reconcile them in order to highlight the current understanding and the gap in knowledge regarding the contribution of SLC26A9 to human lung physiology and CF disease and treatment.

## 1. Introduction

SLC26A9 belongs to the solute carrier family 26 (SLC26), which comprises membrane proteins related to the phylogenetically older SLC26-SulP gene family. As part of the APC superfamily, such proteins operate as electroneutral or electrogenic exchangers of monovalent and divalent anions and as anion channels [1]. The mammalian SLC26 family includes 11 genes (SLC26A1-A11) expressed throughout the body, with a heterogeneous tissue distribution and a variety of functions within a conserved molecular scaffold [1,2]. Three members of the family have been identified as disease-causing genes in different human genetic disorders: SLC26A2 in chondrodysplasias [3], SLC26A3 in chloride-losing diarrhea [4], and SLC26A4 in Pendred syndrome and hereditary deafness (DFNB4) [5,6,7]. Additionally, based on their possible roles in anion transport, some members of the family, including SLC26A4 and SLC26A9, have been linked to cystic fibrosis (CF) as disease modifiers or potential therapeutic targets. CF, which is one of the most common monogenic diseases and affects at least 100,000 individuals worldwide, is caused by mutations in the gene encoding the CFTR anion channel [8]. Although CF is a multi-organ disease, the most severe manifestations are observed in the lung, where the loss of the chloride and bicarbonate secretion mediated by CFTR dehydrates and acidifies the thin layer of fluid covering the airways and impairs the mucociliary clearance and the innate immunity, thus resulting in chronic infection, inflammation, and progressive structural and functional lung damage [9,10,11,12,13,14,15]. The correction of CF abnormalities can be achieved by restoring the function of CFTR with mutation-specific pharmacological treatments [16,17]. However, a potential alternative or synergic approach is to modulate the activity of other channels and transporters in order to stimulate CFTR-independent anion secretion.

In this context, SLC26A9 is under investigation as one of the most interesting targets. Such interest arises from various preliminary observations: (1) it was originally described as a protein expressed in the lung [18]; (2) given the family it belongs to, it presumably mediates anion transport activity; (3) SLC26A9-deficient mice exposed to inflammatory stimuli exhibited a CF-like lung phenotype characterized by airway mucus obstruction [19]; (4) it has already been investigated as a modifier of the phenotype and therapeutic response of CF patients [20,21,22,23]. However, despite these relevant clues, some open issues and remaining controversies, such as the lack of specific pharmacological modulators, the very low SLC26A9 expression in human airways recently reported by single-cell RNA sequencing (scRNA-Seq) studies, and the sometimes-conflicting reports about its relationships with CFTR and CF phenotype, prevent any conclusive understanding of the potentiality of SLC26A9 as a therapeutic target in CF.

In this review, we summarized the various results dealing with SLC26A9 expression at the tissue, cellular, and subcellular levels; its structure and function as anion channel or transporter; its interaction and functional relationships with CFTR; and its role as gene modifier, and tried to reconcile them in order to highlight the current understanding and the gap in knowledge regarding the contribution of SLC26A9 to human lung physiology and CF disease and treatment. 

## 2. SLC26A9 Tissue, Cellular, and Subcellular Localization

### 2.1. SLC26A9 Expression in Human

The original description of SLC26A9 as a lung-expressed protein drew attention to SLC26A9 regarding its possible role in airway physiology and disease [18]. However, following studies focusing on SLC26A9 rarely managed to conclusively confirm and extend such an observation. Figure 1 and Table 1 summarize the major findings regarding SLC26A9 expression in tissues, primary cells, and immortalized cell lines of human origins at the mRNA and protein levels.

As mentioned above, the first observation by Lohi and colleagues detected SLC26A9 by immunohistochemistry with a self-produced antibody (directed against the peptide “ELSLYDSEEDIRSYWD-LEQE”, corresponding to amino acids 758–777 of the human SLC26A9; see Figure 2) on archival specimens of adult human kidney, testis, and lung. SLC26A9 was found to be expressed in the cytoplasm of bronchial and alveolar epithelial cells (Figure 1) [18]. In the same work, SLC26A9 was found endogenously expressed, at the mRNA level, in NCI-H3538 and A549 cell lines (derived from bronchoalveolus and alveolus, respectively) [18].

Besides these findings, not much more effort has been made in the characterization of SLC26A9 protein expression in human airway tissue, until the very recent work by Pinto and colleagues [24]. These authors found some SLC26A9-positive signals by immunofluorescence on native human bronchial tissues (by using a commercially available antibody, NBP2-30425) [24]. SLC26A9 appeared localized in the apical pole of bronchial epithelial cells, possibly overlapping CFTR and the tight junction marker ZO-1, in control tissues, whereas showed cytoplasmic staining in tissues from CF donors carrying F508del-CFTR [24].

Various studies have investigated SLC26A9 expression in human tissues at the mRNA level by RT-PCR, Northern blotting, and scRNA-Seq, indicating the lung, the pancreas (ductal and acinar cells), the prostate, and the stomach (fundus/corpus and antrum region) as the main sites of expression, whereas lower levels have been found in the proximal duodenum (Figure 1) [18,25,26]. Regarding the airway tissues, recent results provided by scRNA-Seq studies seem to indicate a very low expression in airway epithelial cells of the upper airways, including ciliates cells, club cells, and ionocytes, whereas a higher level has been detected in alveolar cells (particularly type 2) [27].

Given its possible role in CF, SLC26A9 expression has been investigated in epithelial cells collected from CF and non-CF individuals and cultured under air–liquid conditions. Different research teams have provided evidence of some SLC26A9 expression at the mRNA level [23,28,29,30]. Sato and colleagues also detected SLC26A9 by immunofluorescence (with the NBP2-30425 antibody) on well-differentiated bronchial epithelia, localizing it at the apical membrane periphery, close to the tight junctions of goblet and ciliated cells [31]. Interestingly, the disruption of the tight junction complexes by the calcium chelating agent EGTA caused both ZO-1 and SLC26A9 immunofluorescence signals to become more diffuse [31].

Concerning the subcellular localization, additional and sometimes controversial data come from studies relying on the heterologous expression of SLC26A9 in Xenopus oocytes and various immortalized cell lines. The transfection of the full-length coding sequence in HEK-293 cells resulted in a high rate of protein expression with the accumulation of intracellular aggregates and almost an absence of SLC26A9 protein at the plasma membrane [32,33]. Retention in intracellular compartments was also the main behavior found in FRT-transfected cells, with only a minor fraction of the protein apparently able to traffic to the cell periphery [34]. Partial trafficking to the cell surface, detected by immunofluorescence or biotinylation approaches, was also reported in Xenopus oocytes injected with the human SLC26A9 mRNA [21], and in CFBE41o^−^- and BHK-transfected cells, particularly if cultured on porous supports [31,35,36]. Interestingly, Dorwart and colleagues found that SLC26A9 surface expression in oocytes and HEK-293 cells was negatively regulated by WNK kinases [37].

A more convincing localization in the plasma membrane of CFBE41o^−^ and HEK-293 cells was observed only by transfecting SLC26A9 constructs carrying a modification of the C-terminal coding sequence, including the deletion of the PDZ or/and the STAS domains (Figure 2) [31,32].

### 2.2. SLC26A9 Expression in Mouse

Other data come from studies carried out on the murine Slc26a9. At the mRNA level, high expression has been detected by RT-PCR and Northern blotting in the stomach, trachea, lung, and duodenum crypt [19,25,38]. Lower levels have been found in the brain, heart, kidney, thymus, spleen, and ovary [39]. In situ hybridization revealed abundant expression in the surface epithelial cells of the stomach [38]. Slc26a9 protein was detected by immunofluorescence labeling by different authors. By using a self-produced anti-Slc26a9 antibody (directed against the mouse synthetic peptide “CDTEFSLYDSEEEGP”, corresponding to human residues 755–778 in Figure 2), a positive signal was found in stomach surface epithelial cells with apical localization and in the body of gastric glands [38]. Similarly, with a second self-produced antibody (directed against the C-terminus sequence “CKQKYLRKQEKRTAIPTQQRK”, corresponding to human residues 565–584), Slc26a9 was found localized in the lung (bronchial epithelial and alveolar cells) and stomach (apical membrane of gastric surface epithelia and intracellular membranes) [39]. Slc26a9 was also found at the apical membrane of principal cells in the medullary collecting duct of the kidney [40].

## 3. SLC26A9 Protein Structure and Function

### 3.1. SLC26A9 Structure

Basic features of the SLC26A9 structure have been initially deduced from the crystal structure of the bacterial homolog SLC26Dg, showing a transmembrane domain (TM) folding such as UraA and NBCe1 [2]. Subsequently, Walter and colleagues gained insight into the SLC26A9 structure by the cryo-electron microscopy (cryo-EM) of a truncated version of the mouse protein [32]. Indeed, in order to improve protein purification, the authors designed a minimal construct where two predicted intrinsically disordered regions were removed from the C-terminal part of the protein (residues P558-V660 of STAS domain and the final 44 residues of the C-terminus P745-L790 containing a PDZ motif) [32]. Such modifications, besides increasing the protein purification rate, also resulted in the higher plasma membrane localization of the truncated SLC26A9 with respect to the full-length murine construct [32]. More recently, Chi and colleagues were able to resolve, again by cryo-EM, the molecular structure of the full-length human protein at an overall resolution of 2.6 Å [33].

SLC26A9 exists as a homodimer. Each protomer can be divided into a TM domain and a cytosolic domain composed, respectively, of 14 transmembrane segments and the N- and C-terminal tails of the protein. As modeled in Figure 2, the small N-terminus connects with the first of 14 transmembrane helices, whereas the C-terminus contains a PDZ domain, and a Sulfate Transporter and Anti-Sigma factor antagonist (STAS) domain linked to TM14. Within the TM domain, different helices assemble to form the flanking core (TM1–4 and TM8–11) and gate (TM5–7 and TM12–14) domains. In the center of each protomer, the TM3 and TM10 shape a half-unwound helix packed with the remaining helix, forming the canonical substrate-binding pocket, as previously determined for UraA and UapA [33].

The C-terminal sequence of SLC26A9 displays unique features and appears folded in the intracellular pocket lined by TM5, TM8, TM10, and TM12. Such a position alters the surface charge from positive to negative. In this way, the C-terminal may reduce the ion accessibility from the cytosolic side, possibly modulating the ion transport kinetics [33]. Accordingly, a C-terminal truncated protein, investigated by single-channel recordings, showed three-times-larger currents than the wild-type protein [33]. 

Moreover, a similar gain of function was obtained by mutating residues within the sequence involved in the interaction between the C-terminus and the TM12-TM5, further highlighting an inhibitory feature of the SLC26A9 C-terminus.

Multiple ions, possibly two Cl^−^ and one Na^+^, were resolved in the high-resolution structure provided by Chi and colleagues [33]. The molecular dynamic simulation and mutagenesis of specific sites indicated the unstable binding of Cl^−^ and Na^+^ to the substrate-binding site, a behavior corresponding to the rapid Cl^−^ flux observed in SLC26A9, and proposed Cl^−^ as being the main ion transported [32,33].

### 3.2. SLC26A9 Transport Properties

The transport properties of SLC26A9 have been investigated by means of various experimental approaches. SLC26A9 was found to be permeable to SO₄²-, Cl^−^, and C₂O₄²^−^ by isotope uptake assays in *Xenopus* oocytes expressing the human protein [18]. Intracellular pH measurements with the BCECF probe in HEK-293-transfected cells found SLC26A9 mediating Cl^−^/HCO_3_^−^ exchange and Cl^−^-independent HCO_3_^−^ extrusion [38]. Subsequently, by isotope uptake and patch-clamp recording in oocytes, HEK-293, and CHO cells, mouse Slc26a9 was associated with three ion transport modes: electrogenic nCl^−^/HCO_3_^−^ exchange, electrogenic Na^+^/nAnion^−^ cotransport, and anion channel [39]. The anion channel activity was further proposed in other studies, showing a constitutive Cl^−^ secretion with linear current–voltage relationship [28,34,37]. Recently, the resolution of the molecular structure of the mouse and human proteins, coupled to structure–function relationships studies, has been a major step forward in the definition of SLC26A9 transport properties, classifying it as a chloride transporter with channel-like activity (i.e., fast kinetics) [32,33].

### 3.3. SLC26A9 Interaction with CFTR

Several studies have investigated the physical and functional interaction between SLC26A9 and CFTR.

A possible direct interaction between SLC26A9 and CFTR, involving the STAS domain of SLC26A9 and the regulatory domain of CFTR, was depicted in *Xenopus* oocytes and resulted in the inhibition of SLC26A9 activity [41]. Many other studies investigated the functional outcomes of SLC26A9 and CFTR co-expression.

Concerning CFTR activity, the co-transfection of SLC26A9 and wt-CFTR in HEK-293 cells resulted in enhanced forskolin-stimulated CFTR-dependent current [28]. Importantly, in immortalized human bronchial epithelial cell lines (CFBE41o^−^ and 16HBE14o^−^), SLC26A9 overexpression and silencing resulted in increased and reduced CFTR expression and function, respectively [24].

Concerning SLC26A9 activity, some studies reported a low contribution to anion transport, either in the presence or in the absence of CFTR [36,42]. Rather, SLC26A9 transfection in HEK-293 resulted in the appearance of constitutive anion currents, both if expressed alone or in case of co-expression with wt-CFTR [28]. A similar behavior was observed with the co-expression of G551D-CFTR, a gating mutation with a normal expression at the plasma membrane [34,35]. Instead, if the co-expression occurred with F508del-CFTR, a trafficking mutation causing endoplasmic-reticulum retention, a noteworthy reduction in SLC26A9 activity was shown in three different models (HEK-293 and BHK cells, and *Xenopus laevis* oocytes) [31,35,36]. Such findings suggest that the two proteins interact early during their intracellular maturation. Therefore, the presence of the trafficking-incompetent F508del-CFTR may prevent SLC26A9 from moving to the cell membrane. This hypothesis was supported by coimmunoprecipitation assays, where SLC26A9 coimmunoprecipitated with both the mature and immature CFTR [35].

Such relationships were further confirmed by studies involving CFTR modulators. In BHK cells transfected with F508del-CFTR and SLC26A9, the treatment with low temperature or the CFTR corrector VX-809 (both maneuvers able to partially rescue F508del-CFTR trafficking) increased the SLC26A9 cell surface expression [31]. Similarly, in human bronchial epithelia from F508del/F508del donors, the rescue of mutant CFTR with VX-809 increased the constitutive chloride currents, possibly due to SLC26A9 [31,35]. Moreover, SLC26A9 overexpression in immortalized human bronchial epithelial cells increased the rescue of F508del-CFTR by correctors [24].

### 3.4. SLC26A9 Contribution to Epithelial Ion Transport

The contribution of SLC26A9 to transepithelial anion secretion has been investigated in different epithelial models. The heterologous expression of SLC26A9 in FRT cells cultured on porous supports increased the baseline anion current measured by short-circuit current recordings [34]. Such activity was abolished under the Cl^−^-free condition and was inhibited by GlyH-101, niflumic acid, and DIDS (which are all commonly used wide-spectrum anion channel blockers), but not by a specific CFTR inhibitor, CFTRinh-172 [28,34]. A constitutive anion secretion not obviously ascribable to other channels, and inhibited by GlyH-101, was also observed in some studies on human bronchial epithelia [28,35]. This constitutive current was affected by tight junction disruption and was absent in bronchial epithelia from F508del homozygous patients, both conditions reported to reduce SLC26A9 expression at the plasma membrane [31,35]. These findings suggest SLC26A9 as the responsible molecular identity, although conclusive evidence is lacking. Moreover, the contribution of such potential SLC26A9-dependent anion secretion to epithelial physiology and disease still needs to be investigated. 

In mice airways, SLC26A9 may play a clearer role, particularly under inflammatory conditions characterized by a Th-2 response. Indeed, IL-13 treatment increased Cl^−^ secretion in the airways of wild-type but not SLC26A9-deficient mice, with the latter thus exhibiting airway mucus obstruction [19]. SLC26A9-deficient mice also showed gastrointestinal manifestations, including decreased gastric acid secretion in a framework of extensive stomach pathology [43].

## 4. SLC26A9 as a Modifier of CF Phenotype and Treatment

Different studies have defined SLC26A9 as a modifier gene of the gastrointestinal tract manifestations of cystic fibrosis. Various SNPs at its 5′ have been found to be associated with meconium ileus, pancreatic damage, immunoreactive trypsinogen at birth, and CF-related diabetes (CFRD) [44,45,46]. A recent study showed that the association with age at onset of CFRD is likely due to co-inherited haplotypes of multiple variants over several regions of SLC26A9, affecting the level of SLC26A9 expression in the pancreas [26].

Nevertheless, the association of SLC26A9 SNPs with lung function is controversial. Several SLC26A9 SNPs have been investigated. For example, the rs7512462 SNP (T > C) occurring in intron 5 is highly debated. The minor allele C was associated with improved lung function in individuals with at least one G551D allele (8.5% increase in FEV1pp for each additional C allele) [20]. In another study, rs7512462*CC was again found to be associated with the amelioration of several lung function parameters (Shwachman–Kulczycki score, FVC, FEV1, FEV1/FVC, FEF50%, and FEF25–75), with respect to the rs7512462*CT [47]. Following studies produced different results. In particular, a decrease of 7.7% in FEV1 was observed for the CC genotype by Corvol and colleagues, and no evidence of the association between rs7512462 genotype and lung disease severity was found by the Survival- adjusted, Kulich-normalized (SaKnorm) lung phenotype in a larger cohort of CF patients carrying at least one G551D allele [48,49]. Moreover, in a meta-analysis of genome-wide studies primarily involving subjects carrying F508del-CFTR mutation, none of the SLC26A9 variants was found to contribute to the lung function heterogeneity [48]. Additionally, no association was shown between the co-inherited CFRD risk haplotypes and lung disease severity [49]. Even considering other parameters within spirometry analyses, such as the age of the first infection with *P. aeruginosa* and residual lung function, SLC26A9 was shown not to be involved in lung function [45].

Additionally, the association between SLC26A9 SNPs and the therapeutic efficacy of CFTR modulators has been the focus of different investigations. The rs7512462 SNP explained 22% to 28% of the response variability to ivacaftor treatment for patients carrying G551D mutation, with improved response associated with the C allele [20,48]. Moreover, clinical data from F508del homozygous patients and the extent of F508del-CFTR function rescued by lumacaftor and ivacaftor in patient-derived nasal epithelial cultures correlated with the rs7512462 SNP [23]. The C allele was associated with increased baseline and rescued CFTR activity, and better clinical outcomes in terms of sweat chloride test and BMI. However, such positive correlations were not observed for the lung function, which resulted in the higher presence of the T allele [23].

## 5. Conclusions and Perspectives

SLC26A9 is emerging as a focus for the development of novel therapeutic strategies for CF disease. Already-approved drugs, based on potentiator and correctors able to rescue the function of mutant CFTR, are showing very promising results in the treatment of several CF patients, particularly those carrying gating mutations and F508del. However, there is a significant fraction of CF patients with “undruggable” mutations. In such cases, as well as for possible adjuvant or synergistic approaches, a strategy based on an alternative target can be considered. In the case of SLC26A9, it could be speculated that the potentiation of its anion transport activity may circumvent the reduced chloride secretion due to the loss of function of CFTR (Figure 3).

However, the knowledge of the role of SLC26A9 in human physiology and CF disease is still in its infancy. Besides different studies seem to indicate a relevant role of this protein in the gastrointestinal tract, the contribution of SLC26A9 to the lung function is still elusive, highlighting the need for further investigations. For example, despite a few studies reporting some SLC26A9 expression in human bronchial airways [18,24], others have failed to find relevant expression or detected SLC26A9 only in a small fraction of cells [27,40,50,51,52]. In addition, SLC26A9, when detected, has often been found expressed in intracellular compartments, with very low expression at the cell membrane, thus questioning its real contribution to transepithelial anion transport [18,24,31]. Therefore, a first step in the understanding of SLC26A9 potentiality as a therapeutic target in CF lung disease should be a better definition of its expression pattern in terms of extent, airway region (e.g., upper versus lower airways, surface epithelia vs. submucosal glands), and cell type, as well as the subcellular localization and the relationships with other channels and transporters. Similarly, a review of the literature on SLC26A9 may suggest the need to gain insight into SLC26A9’s contribution to human airway physiology, possibly combining genetic and pharmacological strategies. For the former, many studies have relied on overexpression and/or silencing approaches but have rarely managed to clearly define the contribution of SLC26A9 to ion transport mechanisms, possibly because of issues that are intrinsic to these methods, such as transfection efficiency and the difference in the constructs and cell models used among the studies. In this regard, the genome editing of human airway (nasal or bronchial) epithelial cells, which are the most CF disease-relevant cell model, in order to ablate or activate SLC26A9 could be a promising approach. Regarding the pharmacological strategy, it should be emphasized that, to date, there is a complete absence of specific modulators. Indeed, studies have usually been performed by using wide-spectrum anion channel blockers, such as GlyH-101, niflumic acid, and DIDS, thus hampering the possible conclusions concerning the contribution of SLC26A9 to ion transport in the airways. Hence, the generation of tools and assays suitable for high-throughput screening approaches to find novel potent and specific modulators of SLC26A9 would be another important focus for future research in this field. To this end, the recent resolution of SLC26A9 molecular architecture coupled to structure–function relationship studies [32,33] has started to shed light on the structural features and regulatory mechanisms underlying SLC26A9 transport characteristics. The identification of a gating mechanism mediated by its C-terminal sequence may suggest ways to modulate its channel-like activity by pharmacological applications [33]. For example, we can highlight the search for small molecules able to alter the interaction between the C-terminal and the intracellular vestibule of the protein in order to enhance the anion transport ability of SLC26A9, as shown by the gain of function effect of C-terminal truncated SLC26A9 constructs [32,33]. On the other hand, other approaches can be envisioned to increase the expression, trafficking, and stability of SLC26A9 protein at the plasma membrane of airway epithelial cells, including, for example, the disruption of the interaction with F508del-CFTR to escape from the related endoplasmic reticulum retention and degradation [35].

## Figures and Tables

**Figure 1 biomolecules-12-00202-f001:**
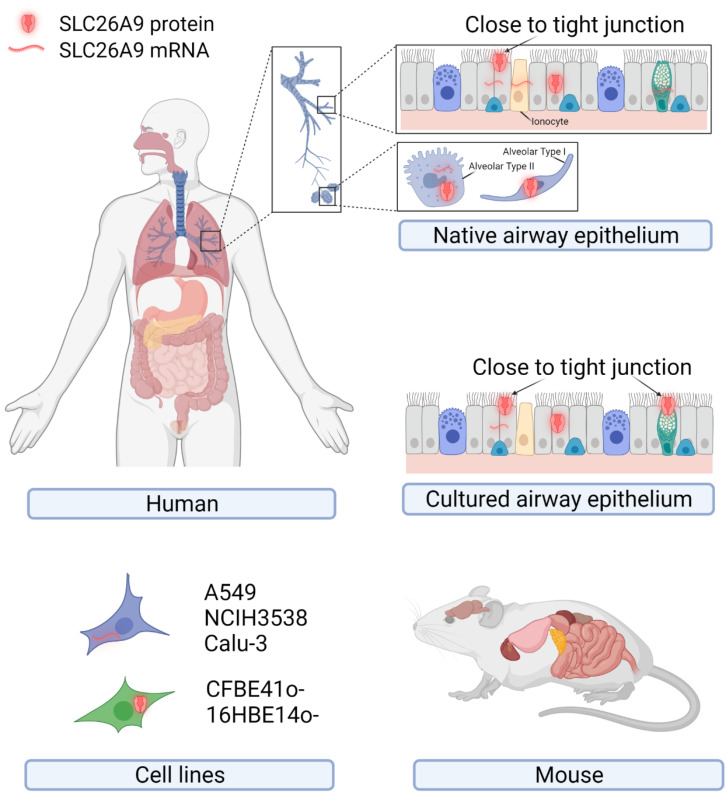
Endogenous expression of SLC26A9 in human tissues, primary and immortalized cell lines of human origin, and mouse tissues. The cartoons show tissues and cells reported to express SLC26A9 at the mRNA and/or protein levels. Details of the putative subcellular localization (intracellular compartments or plasma membrane close to the tight junctions) are shown for the native and cultured airway epithelia. Created with BioRender.com (21 January 2022).

**Figure 2 biomolecules-12-00202-f002:**
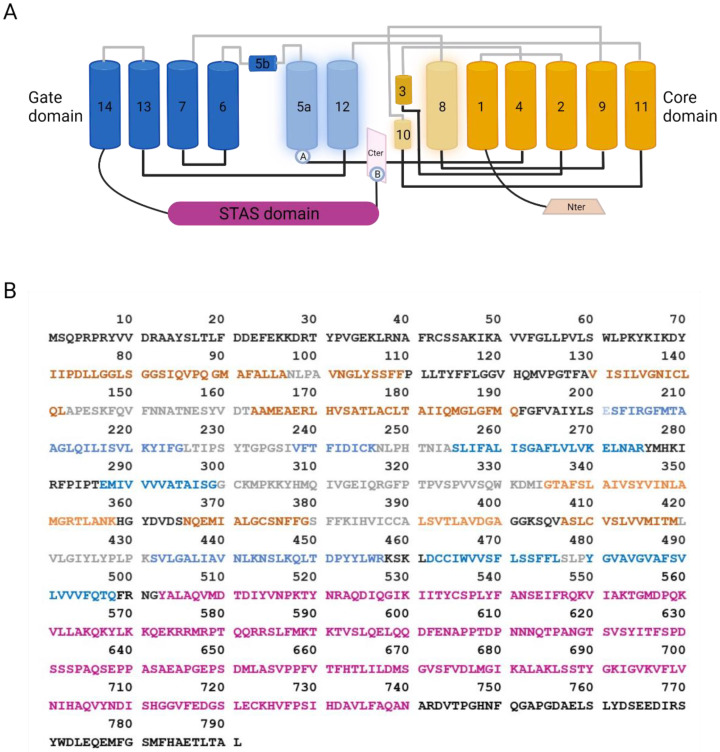
SLC26A9 structure. (**A**) Cartoon showing the topology of SLC26A9, as determined by Chi and colleagues. The gate domain is colored in blue, the core domain is colored in orange, and their participation in the intracellular pocket is highlighted by increased transparency. Extracellular and intracellular loops are colored in grey and black, respectively. The STAS domain is colored in purple, C-terminal in pink, and N-terminal in light orange. Ⓐ: Location of E201K mutation in TM5, which, together with the C-terminal mutation S781A, provided increased currents measured by single-channel recordings. The same results were obtained by mutating different residues visually represented by Ⓑ: E775A/Q776A/F779G/S781G/M782G/F783G and E775A/Q776A. Those mutations can interfere with the interaction between the C-terminal and the TM5 or TM12 [33]. Created with BioRender.com (21 January 2022); (**B**) Amino acid sequence of SLC26A9. Residues are colored as the corresponding domains in (**A**).

**Figure 3 biomolecules-12-00202-f003:**
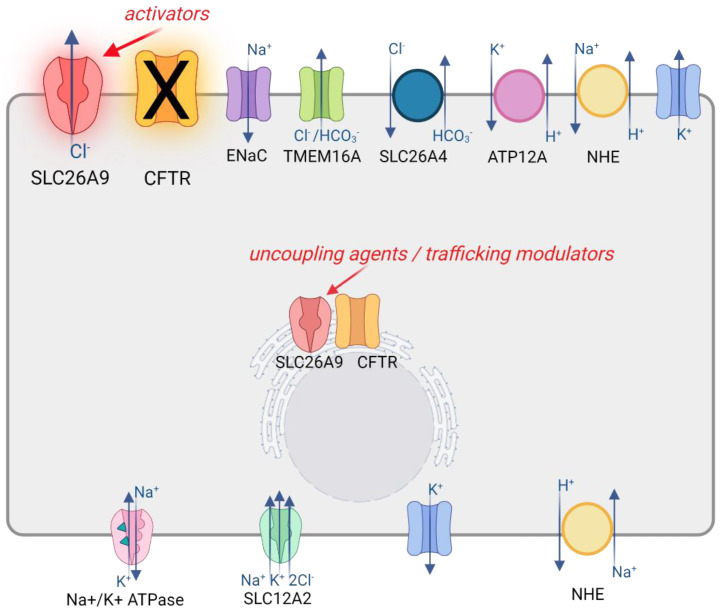
SLC26A9 as a potential therapeutic target in CF. The cartoon shows a simplified airway epithelial cell model with some of the various transporters, channels, and exchangers involved in ion secretion mechanisms. Among these, SLC26A9 could be potentially targeted to circumvent the reduced chloride secretion due to the loss of function of CFTR by searching for small molecules acting as activators or potentiators of its ion transport activity. Additional strategies could involve the finding of trafficking modulators being able to increase SCL26A9 expression at the plasma membrane, possibly acting also as uncoupling agents with respect to the interaction with F508del-CFTR. Created with BioRender.com (21 January 2022).

**Table 1 biomolecules-12-00202-t001:** Endogenous expression of the human SLC26A9. Data from the literature are listed as: “detection method: sample showing expression”. IHC: immunohistochemistry; IF: immunofluorescence.

Tissues		Primary Cells		Cell lines		References
mRNA	Protein	mRNA	Protein	mRNA	Protein	
RT-PCR, Northern blot: lung, pancreas, prostate	IHC: lung bronchial and alveolar cells	-	-	RT-PCR: NCIH3538, A549	-	[18]
-	-	RT-PCR: CF and non-CF bronchial epithelial cells	-	-	-	[28]
real-time PCR: GI tract (mostly stomach)	-	-	-	-	-	[25]
-	-	qRT-PCR: non-CF nasal cells, CF, and non-CF bronchial epithelial cells	-	qRT-PCR: Calu-3	-	[29]
-	-	TaqMan^®^SNP Genotyping system: nasal cells	-	-	-	[23]
-	-	-	IF: non-CF bronchial epithelial cells (goblet and ciliated cells)	-	-	[31]
scRNA-Seq: pancreas (ductal and ductar/acinar cells)	-	-	-	-	-	[26]
-	-	sqRT-PCR: CF and non-CF bronchial epithelial cells	-	-	-	[30]
IF: CF and non-CF lung tissues	-	-	Western blot: nasal epithelial cells	Western blot: Immortalized human bronchial epithelial cells–16HBE14o-, CFBE41o-	-	[24]

## Data Availability

Not applicable.

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
