# Peer review of "SLC26A9 as a Potential Modifier and Therapeutic Target in Cystic Fibrosis Lung Disease"

_biomolecules, 2022, doi:10.3390/biom12020202_

Round 1
Reviewer 1 Report
Giulia Gorrieri et. al, have collected interesting information about SLC26A9 from previously published articles and successfully summarized the existing knowledge. Current review has the attractive information on SLC26A9 from Cryo-EM structure to interaction with other proteins such as CFTR. Article is well presented and is beneficial to researchers and clinicians especially in the CF area. I had no specific comments for the authors except the following typo error and a minor suggestion:
- Line 101: Place a period (.) after … level [23, 29-31]
- Authors are suggested to mention about the initial findings on generation of Slc26a9 deletion mice, decreased gastric acid secretion in the Slc26a9−/− animals [ex: PNAS, 2008 105 (46) 17955-17960)]
Author Response
We thank the Reviewer for her/his very positive comments.
As suggested we have mentioned the findings on the gastrointestinal manifestations of the Slc26a9-deficient mice (line 273-275).
Reviewer 2 Report
This review summarizes the molecular role of SLC26A9 SLC26A9 expression at the tissue, cellular and subcellular levels, its structure and function as anion channel/transporter, its interaction, and functional relationships with CFTR and in CF pathophysiology as a modifier gene.
- A visual scheme/figure that shows different tissues and models where SLC26A9 was detected would be visually informative for readers.
- Line 37, are the authors completely sure about this statement? More common than Familial Hypercholesterolemia? Or then PKU?
- Line 196 There seem to be a typo in electrogenic Na+ /nAnion-
- Line 268/270 Not clear in the CC genotype or the CT genotype was associated with amelioration of phenotype-
- Since the focus of this review is the role of SCL26A9 in CF, a figure summarizing its possible role with CFTR and in CF should be added.
- A lot of groups are interested in measuring SCL26A9 function. More details about protocols for measuring SCL26A9 function will be very valuable for this review.
Overall, this was a very nice review to read, that showed the overall picture of what is known and still to be discovered about SCL26A9, and I recommended that it is published.
Author Response
We thank the Reviewer for her/his positive comments.
We have revised the manuscript according to the Reviewer comments and suggestions. In particular:
- A visual scheme/figure that shows different tissues and models where SLC26A9 was detected would be visually informative for readers.
We thank the Reviewer for this suggestion. In the revised version of the review, there is an additional figure (Figure 1) showing tissues, cells and models reported to endogenously express SLC26A9.
- Line 37, are the authors completely sure about this statement? More common than Familial Hypercholesterolemia? Or then PKU?
We thank the reviewer for this right comment. Accordingly, we modified the text to "one of the most common".
- Line 196 There seem to be a typo in electrogenic Na+ /nAnion-
In this case "n" indicate multiple Anion- (>1), and it is related to the electrogenic transport.
- Line 268/270 Not clear in the CC genotype or the CT genotype was associated with amelioration of phenotype-
This is a matter of debate, with different studies showing different results. However in that point, we are reporting data from the literature indicating that the minor allele C was associated with improved lung function, with 8.5% increase in FEV1 for each additional C allele.
- Since the focus of this review is the role of SCL26A9 in CF, a figure summarizing its possible role with CFTR and in CF should be added.
We thank the Reviewer for this suggestion. An additional figure (Figure 3) about the role of SLC26A9 as a target in CF is now present in the conclusions and perspectives section.
- A lot of groups are interested in measuring SCL26A9 function. More details about protocols for measuring SCL26A9 function will be very valuable for this review
We agree with the reviewer about the relevance of studying SLC26A9 at the functional level. Our review in the revised version contains different references and comments about this focus. In particular, we stressed the need to develop novel tools as, for example, specific pharmacological modulators in order to dissect the SLC26A9 contribution to ion transport mechanisms.